# Unusual Acid Sites in LSX Zeolite: Formation Features and Physico-Chemical Properties

**DOI:** 10.3390/ma16062308

**Published:** 2023-03-13

**Authors:** Aleksandra A. Leonova, Svetlana A. Yashnik, Evgeny A. Paukshtis, Maksim S. Mel’gunov

**Affiliations:** Federal Research Center “Boreskov Institute of Catalysis SB RAS”, Pr. Akad. Lavrentieva 5, 630090 Novosibirsk, Russia

**Keywords:** LSX, surface acidity, adsorption

## Abstract

The advanced approach for the preparation of the NH_4_ form of highly crystalline LSX zeolite under gentle drying conditions (40 °C, membrane pump dynamic vacuum) is discussed. Decationization of this form at moderate temperatures led to the formation of Brønsted acid sites (BASs), whose concentration and strength were characterized by IR spectroscopy. It was found that a maximum concentration of three BASs per unit cell can be achieved at 200 °C prior to the initiation of zeolite structure degradation. The proton affinity of BASs is unusual, and aspires 1240 kJ/mol, which is significantly higher compared to faujasites with higher moduli. The increase in temperature of the heat treatment (up to 300 °C) resulted in thermal decomposition of BASs and the manifestation of amorphous phase with corresponding Lewis acid sites (LASs) as well as terminal Si–OH groups. Both the destruction of BASs and formation of the LAS-containing amorphous phase are the key reasons for the significant decrease in the adsorption capacity in the micropore region revealed for the sample decationized at 300 °C.

## 1. Introduction

Synthetic low-modulus zeolite LSX (structural type FAU) is characterized by its high aluminum content (molar Si/Al ratio ~1.1) [1,2]. Due to the high concentration of extra-framework cations, this material is widely used for adsorption separations [3,4,5,6,7,8,9] and “greenhouse” gas sequestration [10,11,12]. It is effective in the production of ultrapure oxygen for medical purposes [13,14,15]. It also has prospects for the adsorption storage of H_2_ [16,17], and can also be used in low-temperature catalytic transformations of biomass [18].

Important characteristics relevant to any zeolite catalyst are nature, strength and concentration of active surface sites, including Brønsted acid sies (BASs). Partially or completely decationized LSX zeolite can be characterized by two unique features. Firstly, acid sites of assumedly high concentrations are accessible to relatively large substrate molecules with a limiting size of 7.4 Å (e.g., CO, CH_4_, CO_2_, N_2_O, neopentane, trimethylbenzene, *o*-xylene, *p*-xylene, etc.) [19]; secondly, all acid sites are almost identical in structure, being supposedly homogeneously distributed over the surface. However, studies on the decationized forms of LSX are extremely limited due to the considerable preparation difficulties [20]. BASs in zeolites are formed as a result of the charge-compensating NH_4_^+^ cation thermolysis that results in partially or completely decationized (H^+^-) forms of zeolites having characteristic groups of –Si–O(H)–Al– [21,22]. These sites are characterized by temperature-programmed desorption of ammonia (NH_3_ TPD) [23] and infrared (IR) spectroscopy of CO [22,24,25] or pyridine molecules [25,26,27,28]. BASs in high-modulus Y and USY faujasites catalyze reactions for oil refining and the production of branched organic isomers [29,30,31,32,33,34,35,36,37,38]. In industry, high-modulus zeolites are converted to decationized forms that are often subjected to additional treatment in order to develop secondary porosity for diminution of diffusion restrictions [32,39,40,41]. This procedure is carried out at high temperatures in the presence of water vapor that usually results in local destruction of –Si–O(H)–Al– groups [39,42,43,44]. The so-called controlled decationization is used to preserve the crystalline structure and the simultaneous formation of BASs in X zeolites [45,46]. However, low-modulus LSX zeolite results in complete amorphization of the crystalline structure under the same conditions due to the high concentration of –Si–O(H)–Al–groups [20,41]. A small number of studies have been devoted to application of this approach for type-X and LSX zeolites [41,45,46,47,48,49], so far. We did not find any systematic analysis of concentration and localization of BASs in such materials.

As we already mentioned, presently, one of the most important applications for low-silica zeolites is the separation of gases, particularly the adsorption separation of mixtures O_2_/N_2_, H_2_/CO, CO_2_/N_2_O, etc. by using PSA (pressure swing adsorption) or VSA (vacuum swing adsorption) processes [11,12]. E.g., commonly used for these purposes are low-silica zeolites with high lithium content, which can be prepared by means of multiply repeated cation exchange. The high price of lithium leads to the necessity of reductions in lithium loss [50,51]. The NH_4_ form of LSX zeolite may be used as an intermediate for the preparation of the Li form, with lithium ions located in the most accessible positions SII and SIII, which further motivates the study of the NH_4_ form of LSX zeolite in this work.

In this paper, we propose an approach to obtain the highly crystalline NH_4_ form of LSX zeolite. The main textural parameters were monitored during its controlled decationization to the protonated form. The nature and concentration of surface acidic sites, including BASs, were characterized by TPD of ammonia, in situ IR spectroscopy and IR spectroscopy of adsorbed pyridine and CO molecules. For the estimation of adsorption selectivity [52,53], adsorption isotherms of pure CO_2_ and N_2_O were measured and analyzed.

## 2. Materials and Methods

LSX zeolite in Na,K form, synthesized by a well-known technique from kaolin [2], was used as the starting material. Ion exchange treatment was carried out in 1 M aqueous solution of ammonia chloride (qualification “reagent grade”) with a solution-to-zeolite ratio (wt./wt.) of 80:3 at room temperature for 3 h under permanent stirring. Then, the sample was filtered, and washed with a three-fold excess amount of distilled water until no chloride ion reactions were observed. Modified and initial samples were dried at 40 °C under reduced pressure provided by a membrane vacuum pump (P ~ 1–2 kPa) for 16 h. The samples were denoted as follows: ***ini***—starting material, ***am***—ion-exchanged sample dried under vacuum at 40 °C. Prior to or during different measurements, the samples were subjected to additional heat treatments under various conditions. The conditions of the thermal treatments are denoted in the description of the corresponding methods of analysis.

The chemical composition of samples was determined by atomic emission spectroscopy with inductively coupled plasma (Optima 4300 DV, Perkin Elmer, Waltham, MA, USA). The samples were not subjected to additional heat treatment before this analysis.

The crystallinity of samples was evaluated from powder diffraction patterns obtained using an ARL X’TRA (Thermo Scientific, Waltham, MA, USA) diffractometer. The range of 2θ was set to 4–40°; λ = 1.54181 (CuKα). The degree of crystallinity (γ) for all samples was calculated according to the ASTM D3906-03 standard [54] accounting for 8 reflexes, which is typical for FAU structures. Starting material ***ini*** was assumed as a material with 100% crystallinity. The samples were not subjected to additional heat treatment before this analysis.

Textural and adsorption characteristics of samples were measured by means of a Quadrasorb evo (Quantachrome Instruments, Boynton Beach, FL, USA) instrument. Before the measurements of adsorption isotherms, samples were treated under dynamic vacuum (P ~ 5–10 Pa) at 200 °C and 300 °C (heating rate 100 °C/h) for 3 h. The corresponding samples were designated as ***ini*-200**/***ini*-300** and ***am*-200**/***am*-300**. Additionally, the ***am*** sample was treated in air at 300 °C (heating rate was approximately 100 °C/min) followed by vacuum heat treatment at 300 °C (heating rate 100 °C/h). The corresponding sample was designated as ***am*-300c**. Finally, sample ***am*-300vac-300c** was prepared as follows: vacuum heat treatment at 300 °C (heating rate 100 °C/h) of the ***am*** sample was followed by treatment in air at 300 °C (heating rate was approximately 100 °C/min). The preparation order for these samples is shown in detail in Figure 1. The following textural characteristics were calculated from the obtained N_2_ adsorption isotherms (77 K): the specific surface area (**A_s_**) was calculated according to the modified BET method [55]; the total pore volume (**V_Ʃ_**) was estimated at p/p_0_ ~ 0.995 according to the Gurvich rule [56]; and the micropore volume (**V_mic_**), the mesopore volume (**V_meso_**) and the mean pore diameter (**D**) were calculated from the pore size distributions, obtained from adsorption branches of the isotherms by a method based on the simulation of adsorption in silicate’s cylindrical pores within the frameworks of the nonlocal density functional theory (NLDFT) implemented in Quantachrome’s software (ASiQwin, v.3.0) [57]. For estimating the static adsorption selectivity, adsorption isotherms of pure CO_2_ and N_2_O at 0 °C on samples ***ini*-200** and ***am*-200** were measured. Adsorption selectivity [53] was calculated by means of IAST++ software (Program Version 1.0.1) assuming a 50/50 *v*/*v* gas mixture composition.

NH_3_ TPD experiments for the ***ini*-200**, ***am*-200** and ***am*-300** samples were carried out in a flow reactor with on-line signal registration by a thermal conductivity detector. The initial (Na,K form) and ammonia-modified LSX samples were fractionated (size = 0.25–0.5 mm). Then, 100 mg of the sample was mixed with 100 mg of quartz of a similar particle size, loaded into the reactor and pretreated at 200 or 300 °C in a helium flow (50 cm^3^/min) for 2 h to in situ prepare the ***ini*-200**, ***am*-200** and ***am*-300** samples and remove from them any adsorbed water. Here, the heating rate was 100 °C/h. The samples were then cooled to 75 °C, after which, NH_3_ was adsorbed by purging the sample with a mixture of ammonia (0.35 vol.%) in helium for 1 h. When adsorption was finished, the samples were purged with helium (50 cm^3^/min) for 1 h to remove physically adsorbed ammonia. Finally, the samples were cooled down to room temperature under helium flow. The NH_3_ TPD profiles were recorded by passing helium through the samples at a rate of 30 cm^3^/min accompanied with heating from 25 to 600 °C at a heating rate of 10 °C/min. The total surface acidity of samples was estimated from the amount of desorbed ammonia molecules, assuming monodentate ammonia adsorption.

Decationization of the ***am*** sample was studied by in situ IR spectroscopy using a Varian Scimitar 1000 (Varian, Palo Alto, CA, USA) instrument in the wavenumber range of 2400–4000 cm^−1^, in a CaF_2_ cell under vacuum (1 Pa). Transmission mode of IR spectroscopy was applied. The wafer was prepared by pressing (pelletized) of the sample powder onto thin self-supported discs with densities of about 20 mg/cm^2^. First, the IR spectrum of the samples (dried at 40 °C) was recorded at room temperature, then the temperature was increased from 150 to 350 °C, and the IR spectra were registered in increments of 50 °C.

BASs and LASs on the surface of the ***am*** sample were studied by IR spectroscopy of adsorbed pyridine and CO molecules. In both cases, the IR spectra of the sample were recorded on a Shimadzu-8300 Fourier spectrometer (Shimadzu Scientific Instrument, Kyoto, Japan) with 4 cm^−1^ optical resolution. Prior to measurements, the probe with a weight of 35–50 mg was pelletized, placed in the measuring cell and calcined in situ under vacuum (<1 Pa) at 200 °C or 300 °C for 1 h (heating rate = 200 °C/h). These conditions were analogous to the preparation of ***am*-200** and ***am*-300** samples. Before pyridine and CO adsorption, the IR spectra of the heat-treated sample were registered.

Pyridine adsorption was studied by injecting 5 µL of liquid pyridine onto the sample at 150 °C and, afterwards, the sample was kept at this temperature for 15 min. Excess pyridine was removed under vacuum at 150 °C for 40 min. The strength of BASs (measured as proton affinity, PA) was calculated according to Equation (1), accounting for the gravity center of the bands of stretching vibrations of the N–H bond of the pyridinium ion (ν_cg_) in the spectrum of the studied zeolite sample and the undisturbed N–H bond (3400 cm^−1^) as:PA [kJ/mol] = (Log(3400 − ν_cg_) − 0.115)/0.0023.(1)

The BAS concentration in the sample was measured through the integral intensity of the band at 1540 cm^−1^ from pyridinium ions using an integral absorption coefficient of 3 cm/μmol taken from [58].

Adsorption of CO on the ***am*-300** sample was carried out at −196 °C, by the stepwise injection of gas, to reach pressures of 0.1, 0.4, 0.9, 1.5, 2.5 and 10 torr. The total concentration of BASs determined from CO adsorption was measured from the intensity of the band at 2163–2165 cm^−1^, using a coefficient of 2.6 cm/μmol [59]. BAS strength was evaluated as follows [60]:*PA* [kJ/mol] = 1390 − Log(Δν_OH_^CO^/90)/0.00226.(2)

Here, *PA* is the proton affinity; Δν_OH_^CO^ is the shift of the OH group band due to the formation of a hydrogen bond with the adsorbed CO molecules; and 1390 kJ/mol and 90 cm^−1^ are the same characteristics of the surface OH groups of SiO_2_ (aerosil).

## 3. Results and Discussion

Elemental compositions of the ***ini*** and ***am*** samples recalculated per unit cell (UC) (from AES ICP data) are shown in Table 1. One can observe that more than 50% of alkali metal ions were exchanged with ammonia. The Si/Al atomic ratio in both samples is equal. This can indicate the absence of dealumination of LSX zeolite after ion exchange and vacuum treatment at 40 °C, which usually occurs with this zeolite after less gentle heat treatment.

Dealumination of low-silica zeolites is accompanied by the destruction of the crystal structure, according to the literature [20,39,41,42,43,44]. A constant Si/Al ratio assumes that one should not expect any noticeable decrease in the intensity of reflexes or the emergence of an amorphous halo on powder X-ray diffraction patterns. The samples ***ini*** and ***am*** are characterized (Figure 2) by high intensities of XRD patterns and individual reflexes; there is no halo. Positions of reflexes characteristic for the FAU structure, namely (111), (220), (331), (533), (553) and (715), do not change. The decrease in the calculated value of crystallinity can be explained by the change (redistribution) observed in the intensities of the reflexes due to the change in the cationic composition, as is discussed elsewhere [61,62,63,64,65].

N_2_ adsorption isotherms at −196 °C for samples ***ini*-200**, ***ini*-300**, ***am*-200** and ***am*-300** are shown in Figure 3. The samples ***ini*-200** and ***ini*-300** are characterized as type Ia isotherms with insignificant type-H4 hysteresis loops according to the IUPAC classification [66]. This corresponds to a predominantly microporous structure and is consistent with data found in the literature [1,2]. The modified samples ***am*-200** and ***am*-300** are also characterized by high adsorption capacity over the entire pressure range. The enhanced slope of the isotherms of modified samples within a 0.05 < p/p_0_ < 0.2 region fit them to the Ib type, which implies a wider pore size distribution and even the presence of small mesopores of about 2.5 nm in size. However, the preservation of the microporous structure is achieved by using mild conditions of drying and vacuum treatment, under which there is no simultaneous thermal decomposition of NH_4_^+^ ions, and water desorption or cavitation within the crystal structure [67]. However, the increase in the temperature from 200 °C to 300 °C affects textural properties of the modified sample ***am*** more significantly compared to those of the ***ini*** sample. While N_2_ adsorption isotherms for ***ini*-200** and ***ini*-300** are almost similar, the isotherms for ***am*-200** and ***am*-300** differ in adsorption capacity. This fact suggests that the stability of the microporous structure in the modified sample decreases in the range of 200–300 °C.

To explore the stability of the ***am*** sample, we performed a sequence of thermal treatment both with vacuum degassing and in the air. Figure 4a shows the N_2_ adsorption isotherms at −196 °C for samples ***am*-300** (vacuum), ***am*-300c** (air) and ***am*-300vac-300c** (vacuum followed by air treatment). Pore size distributions for samples ***ini***, ***am*-300**, ***am*-300c** and ***am*-300vac-300c** are shown in Figure 4b. Samples ***ini*-300** and ***am*-300** contain micropores 1.1–1.2 nm in size, characteristic of the structure of faujasite [68,69]. Samples ***am*-300** and ***am*-300vac-300c** have pores in the range of 2–5 nm. Sample ***am*-300c** has no pores around 1.1 nm, but has mesopores in the range of 3–10 nm. Particular textural parameters calculated from the adsorption isotherms are shown in Table 2. All samples, except ***am*-300c**, are characterized by a high specific surface area, consist mainly of micropores and mesopore volume does not exceed 10% of the total pore volume. The ***am*-300c** sample has a mixed micro–mesoporous structure; its fraction of mesopores exceeds 70%. The nature of micropores seems to be different from the FAU structure, since no 1–1.2 nm micropores are observed in the distribution.

According to the obtained data, the thermal treatment of the ***am*** sample in air results in a dramatic change in the porous structure. The most significant change occurs when direct thermal treatment at 300 °C in the air is applied. The N_2_ adsorption isotherm for the ***am*-300c** sample belongs to type IVa, with an intermediate hysteresis type between H1 and H2, indicating transformation of the predominantly microporous material to a sample with a mainly mesoporous structure. The size of the mesopores lies within the 3–10 nm range. Intermediate vacuum treatment in the case of the ***am*-300vac-300c** sample prevents formation of mesopores of this width. The resulting size lies between 2 and 5 nm and corresponds to one to two unit cells. It is possible that we observe successive destruction of micropores due to the action of water remaining in zeolite pores after drying at 40 °C. Intermediate vacuum treatment removes the remaining water and, thus, can be considered milder compared to direct treatment in the air, and results in less intensive micropore structure degradation. On the molecular level, this can be interpreted as follows: Thermal decomposition of ammonium ions practically does not occur during vacuum drying at 40 °C, so the ***am*** sample contains a high concentration of –Si–O(NH_4_)–Al– groups. The air, under any conditions, contains moisture accompanied by the residual moisture in the sample, so in the case of ***am*-300c**, –Si–O(NH_4_)–Al– groups suffer from the simultaneous action of water vapor and the thermal decomposition of ammonium ions. This promotes the destruction of –Si–O(NH_4_)–Al– groups, and successive degradation of the crystal structure, accompanied by a simultaneous decrease in the micropore volume, and the formation of mesopores. Preliminary successive vacuum drying at 40 °C, and then at 300 °C (as it was done in the case of the ***am*-300vac-300c** sample), separates water removal and thermal decomposition of ammonium cations in time. This makes the processing milder, and noticeably preserves microporosity as a result.

The destruction of the ***am*-300c** sample proceeds through the formation of –Si–O(H)–Al– groups as a result of the thermal decomposition of ammonium ions in the presence of water. These groups, being BASs, are visible through IR spectroscopy. Figure 5 shows the IR spectra taken in situ for the ***am*** sample during successive heating under dynamic vacuum from room temperature to 350 °C (the spectra are shifted by 0.5 units of absorbance).

There are no significant features of spectra below 200 °C, which is due to the presence of physically adsorbed water within the porous structure. When the sample is heated up to 200 °C, the absorbance band at 3650 cm^−1^ appears in the spectrum. This band corresponds to bridged OH groups localized in supercages [70,71,72,73,74,75,76]. This is consistent with N_2_ adsorption; the ***am*-200** sample is characterized by the largest adsorption capacity due to the replacement of relatively large and heavy alkaline cations with small and light protons in the SIII positions. An increase in the temperature above 200 °C results in the manifestation of the second band at 3550 cm^−1^, which corresponds to –Si–O(H)–Al– groups in sodalite cages [70,71,72,73,74,75,76,77,78,79,80,81].

The in situ information on the proton sites formed in the ***am*** sample was supplemented by IR spectroscopy data, confirming the presence of residual NH_4_^+^ groups in ***am*-200** and ***am*-300** samples.

Figure 6 shows the IR spectra of ***am*-200** and ***am*-300** samples before pyridine or CO adsorption. As discussed above, OH groups bands at 3580 cm^−1^ (LF band) and 3657 cm^−1^ (HF band) belong to bridging OH groups localized in sodalite cages and large cavities [70,71,72,82] of type-X zeolites, respectively. In addition to stretching vibration bands of OH groups, there are absorbance bands at 1450, 1680 and 2925, 3090 and 3250 cm^−1^. These are caused by the deformation (symmetrical and asymmetrical) and stretching vibrations of ammonium ions. Such bands are typical of the NH_4_^+^ ions localized in various solid acids [83,84,85]. Thus, they indicate residual –Si–O(NH_4_^+^)–Al– groups in both ammonia-modified samples. The intensity of the 1450 cm^−1^ band in the ***am*-300** sample is about four times lower compared to that for the ***am*-200** sample (Figure 6, Curves 1 and 2), which proves the sequential transformation of ammonium ions to proton sites with temperature increase.

The concentration, strength and thermal stability of acidic sites were estimated from NH_3_ TPD experiments (Figure 7). The discussed IR spectroscopy data were used to interpret the NH_3_ TPD. The obtained data show that the integral intensities of the TPD curve (in other words, the total concentration of desorbed ammonia molecules) in ***am*-200** and ***am*-300** samples are 2.9 and 1.7 times larger compared to that of the ***ini*** sample, respectively. The ***ini*** sample was used as a reference with no acid sites. The shape of the desorption profile for ***am*-200** is consistent with that for fully decationized zeolite HY [86,87,88]. The NH_3_ TPD profiles of all samples studied contain an intensive narrow maximum at ~200 °C. The more intense the peak of weakly adsorbed ammonia, the higher the concentration of NH_4_^+^ in the sample. This trend correlates with the 1450 cm^−1^ absorbance band observed in IR spectra of the ***am*-200** and ***am*-300** samples (Figure 6, Curves 1 and 2). On the other hand, the NH_4_^+^ cation is absent in the ***ini*** sample. The low-temperature peak of ammonia desorption is believed to not be directly related to the proton acidity of the samples [87,89], for example, for the ***ini*** sample and other Na,K-containing zeolites [87,89]; however, there is no unambiguous attribution in the literature. This corresponds to weak adsorbed ammonia, formed probably due to the interaction of ammonia molecules with NH_4_^+^ [90] and with residual water in the zeolite pores [47,87] or adsorbed on LASs located near OH groups [87,89]. Sometimes, the low-temperature peak is associated with the desorption of ammonia from terminal Si–OH groups [87].

In addition, NH_3_ TPD profiles of the modified samples are characterized by strongly adsorbed ammonia desorbed at higher temperatures: 285–315, 355 and 470 °C. It is believed that the higher the temperature of ammonia desorption, the stronger the site from which ammonia is desorbed. There is the wide shoulder at 285–315 °C, which corresponds to medium-strong sites for the ***am*-200** sample. Taking into account the IR absorbance bands at 3657 and 3580 cm^−1^ (Figure 6), these sites can be bridging OH groups in large cavities and sodalite cages of LSX. Its intensity in terms of the concentration of ammonia molecules is 213 μmol/g of the sample. The intensity of the shoulder is much lower in the case of the ***am*-300** sample (about 15 μmol/g of the sample), but the small maximum at 470 °C is observed in this case. Strong acidity of the latter could be related to hydroxy groups in the composition of amorphous extra-framework aluminosilicate [23] or Al_ex_–OH [23,88,90] fragments. At elevated temperatures, a portion of the bridging proton groups are removed, which is accompanied by the extraction of Al^3+^ cations from the zeolite framework, the formation of Lewis acid sites and terminal Si–OH groups, as well as the loss of crystallinity [23] as shown by XRD. The slow desorption of ammonia from ***ini*** and ***am*-300** samples in the temperature range of 250 to 360 °C is associated with a change in the acidity of their proton sites (Figure 7). The acidity increases with a decrease in total concentration of BASs [87,89] and an increase in the proportion of extra-framework Al^3+^ [23], but it decreases in the presence of alkaline cations [23,88]. The ammonia desorption at temperatures above 500 °C is usually associated [23] with irreversible ammination of extra-framework Al–OH and Si–OH groups during NH_3_ TPD. This process is enhanced for the ***am*-200** sample, as it adsorbs more ammonia during its pretreatment (Figure 7). Thus, concluding the analysis of NH_3_ TPD, the BAS content in the ***am*-200** sample does not exceed three centers per unit cell; calculations show that 1 g of zeolite contains 74 μmol of UCs with composition Na_32_K_12_(NH_4_)_52_Al_96_Si_96_O_384_.

In addition to the spectra of ***am*-200** and ***am*-300** samples, Figure 6 illustrates difference IR spectra (***am*-200-ads** and ***am*-300-ads**) obtained by subtracting spectra of samples before pyridine adsorption from the corresponding spectra taken after its adsorption at 150 °C. Difference spectra reveal bands at 1490, 1540 and 1637 cm^−1^, which correspond to pyridinium ions [91,92] and are an indicator of the proton sites in the sample under study. The intensity of the band at 1540 cm^−1^ is usually used to measure the concentration of BASs [91,92]. Gould and coworkers [92] estimated the value of the integral absorption coefficient in the range of 1.98–2.98 cm/μmol to calculate the concentration of BASs; we accepted this coefficient equal to 3 cm/μmol [58]. Concentrations of pyridinium ions were found to be 230 and 135 μmol/g for ***am*-200** and ***am*-300** samples, respectively. These correspond to three and two pyridinium ions (or BASs) per zeolite UC. Negative peaks appear on difference spectra at 3580 and 3657 cm^−1^, which specify the loss of OH groups due to their interaction with pyridine. It is worth noting that a part of ammonium ions remain intact in the ***am*-200** sample, since a negative peak is observed in the IR spectrum at 1450 cm^−1^ (cf. Curves 1 and 3). The half-width of this band is close to the peak of pyridinium ions at 1450 cm^−1^. This corresponds to the substitution of ammonium cations for pyridinium cations. After heat treatment of zeolite under vacuum at 300 °C (***am*-300**), this trend is not observed.

Additionally, a wide complex absorption band with maxima at 2100, 2200 and 2450 cm^−1^ is observed. It corresponds to the NH stretching vibrations of the pyridinium ions attached by hydrogen bonds to the acid residues on the surface. The complexity of this band is due to the Fermi resonance inside the N–H–O fragment [58,93,94,95].

The acid strength of BASs was calculated using Equation (1); based on the position of the center of gravity of the band of stretching vibrations of the N–H bond (2340 cm^−1^) in the pyridinium ion (ν_cg_), we evaluated it as 1260 kJ/mol.

The weak band at about 1453 cm^−1^ is also observed on the spectrum of the ***am*-300** sample. This band corresponds to pyridine complexes associated with LASs [25,26]. Their concentration is only 10 μmol/g or less than 1 LAS per UC. These data are consistent with NH_3_ TPD for ***am*-300**. Meanwhile, no LASs were observed in the ***am*-200** sample according to both methods.

Figure 8 shows the IR spectra of CO adsorbed on the ***am*-300** sample. Two bands are observed in the spectra: the band of weak intensity at ~2215 cm^−1^ corresponds to CO complexes with LASs, and the main band with a maximum at 2165–2163 cm^−1^ is related to CO complexes with BASs [24,84,96,97,98]. The intensity of the first band increases with CO pressure up to 2.5 torr; the second band reaches a maximum intensity under 10 torr. The concentrations of acid sites measured from the intensity of the corresponding bands are 7 and 140 μmol/g for LASs and BASs, respectively. These values are close to those measured from the spectra of adsorbed pyridine (10 and 135 μmol/g).

Figure 9 shows the difference spectrum obtained by subtracting the spectrum after adsorption of CO on the ***am*-300** sample at 77 K and p = 10 Torr (Figure 8) from the spectrum of the ***am*-300** sample before adsorption (Figure 6, Curve 2). It is worth noting that the intensity of bands at 3575 and 3665 cm^−1^ decreases, and the broad band of H-bounded OH groups appears in the range of 3200–3800 cm^−1^ with a maximum intensity at 3466 cm^−1^. According to Equation (2), the shift of 200 cm^−1^ (compared to the HF band, 3665 cm^−1^) corresponds to the strength of acid sites in the proton affinity scale of 1240 kJ/mol, which is significantly less than the shift of 250–300 cm^−1^ which is characteristic for HY [97,99,100,101]. This characterizes LSX zeolite in partially decationized form as a material with less strong Brønsted acidity, which corresponds to a smaller Si/Al ratio [102,103,104,105,106]. This value is also close to the PA of the sites determined from the IR spectra of pyridine (1260 kJ/mol).

Undoubtedly, a comparison of the obtained data on H-LSX acidity with those known for faujasites with different Si/Al ratios was of particular interest. Figure 10 shows the values of the proton affinity (a) and concentration of BASs (b) depending on Si/Al ratio. There is the increase in the PA with aluminum content in faujasites, and our current data follow the trend. This trend corresponds to the next nearest neighbors (NNN) model [107,108,109], which assumes the acid strength of BASs increases with the content of Si neighboring Al. Otherwise, the isolated –Si–O(H)–Al– groups have stronger acidity compared to associated Al–(SiO(H))_4_ clusters. Unfortunately, poor stability of the associated Al–(SiO(H))_4_ clusters results in a significant decrease in the BAS concentration compared to the assumedly possible accounting concentration of Al in LSX. According to our data, the concentration of BASs in LSX faujasite is greater than that for Y zeolites with an intermediate module of ~10–30 (Figure 10b).

High PA, compared to Y zeolites, accompanied with low BAS concentration and relatively low hydrothermal stability of H-LSX do not allow for the consideration of this material as a prospective catalyst. However, extra-framework protons and ammonia cations definitely rearrange surface charges. This is manifested in the adsorption of polar molecules. Figure 11 shows adsorption isotherms of pure CO_2_ and N_2_O at 0 °C and the corresponding adsorption selectivity (S_N2O/CO2_) for samples ***ini*-200** and ***am*-200** within the range of pressure ~0–100 kPa. The selectivity was calculated by means of IAST++ software in assumption of a 50/50 *v*/*v* gas mixture composition.

Adsorption uptake of CO_2_ is higher compared to N_2_O for the ***ini*-200** sample in the whole pressure range, which is in consistence with the literature [12,118]. Adsorption uptake for both gases for the ***am*-200** sample is lower (below 50 kPa) and higher (above 80 kPa) compared to ***ini*-200**. In contrast to the ***ini*-200** sample, adsorption isotherms for both gases over ***am*****-200** sample almost coincide. This is reflected in the behavior of the IAST selectivity, which lies within 2.5–3.0 for the ***ini*-200** sample, and reaches 1.2 for the ***am*-200** sample. For example, this favors simultaneous removal of CO_2_ and N_2_O from the polluted air over a single adsorbent.

## 4. Conclusions

This work demonstrates the possibility of preparation of the highly crystallized ammonium-containing form of LSX zeolite. Thermal decomposition of ammonia cations results in the formation of Brønsted acid sites as is confirmed by the NH_3_ TPD and IR spectroscopy of adsorbed pyridine and CO molecules. The concentration of BASs is about three per unit cell—relatively weak compared to BASs in H-Y zeolites (proton affinity is 1240 kJ/mol against 1150–1190 kJ/mol [33,92,94], correspondingly) which is unusual for zeolites, but in a good agreement with next nearest neighbors (NNN) model. The content of BASs in H-LSX decreases with temperature of preliminary vacuum heat treatment in the range 200–300 °C due to their destruction and formation of Lewis acid sites. There is also a tendency to increase the number of BASs in sodalite cages. This fact is in good agreement with the data of N_2_ adsorption: the adsorption capacity decreases with the number of –Si–O(H)–Al– groups (BAS), accompanied by the formation of an amorphous phase consisting of LASs and extra-framework –Si–OH and –Al–OH species, demonstrating porous structure degradation.

## Figures and Tables

**Figure 1 materials-16-02308-f001:**
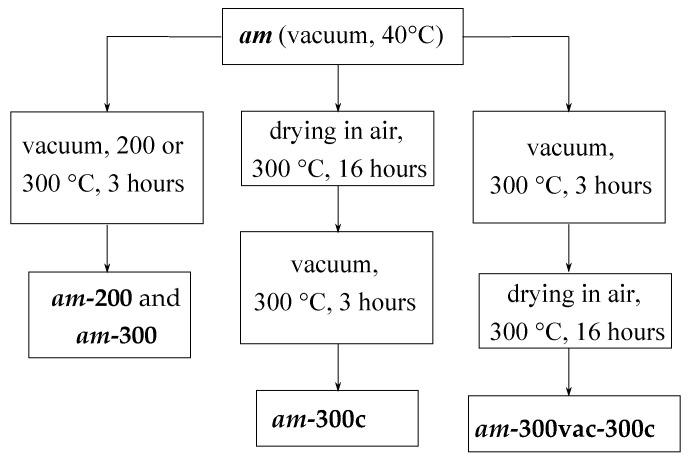
The sequence of heat treatments of samples before N_2_ adsorption measurements.

**Figure 2 materials-16-02308-f002:**
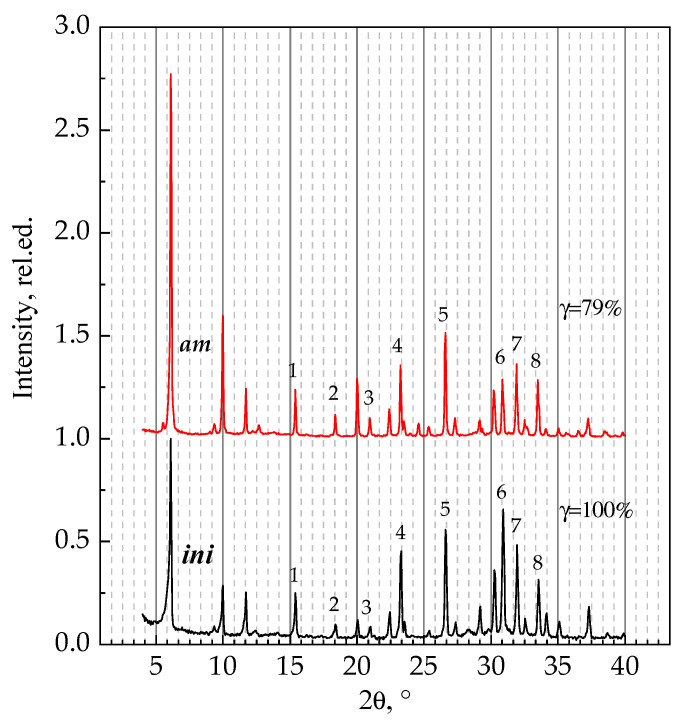
Powder X-ray diffraction patterns of the ***ini*** and ***am*** samples. Numbers correspond to reflexes characteristic for the FAU structure, namely (331)—1, (511), (333)—2, (440)—3, (533)—4, (642)—5, (822), (660)—6, (555), (751)—7, (664)—8.

**Figure 3 materials-16-02308-f003:**
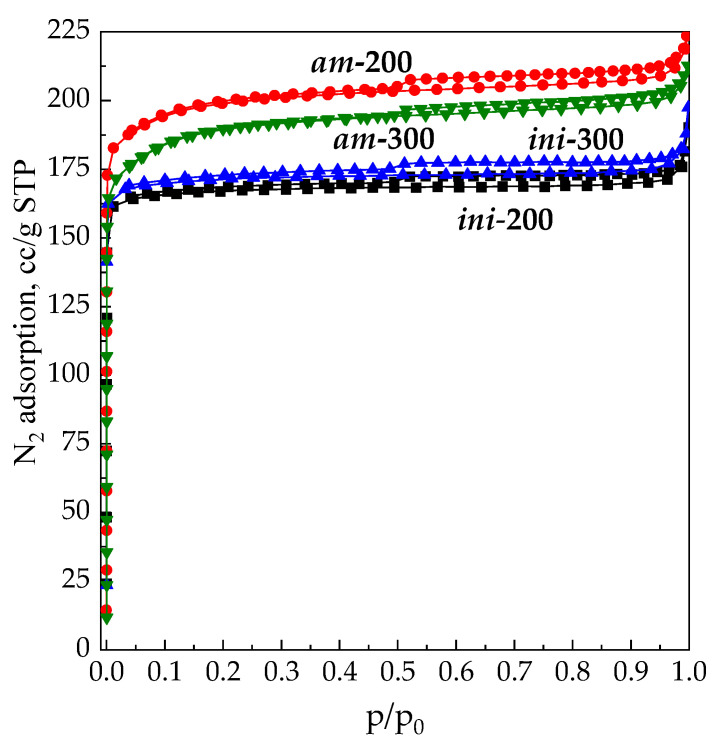
N_2_ (−196 °C) adsorption isotherms on the samples ***ini*-200**/***ini*-300** and ***am*-200**/***am*-300**.

**Figure 4 materials-16-02308-f004:**
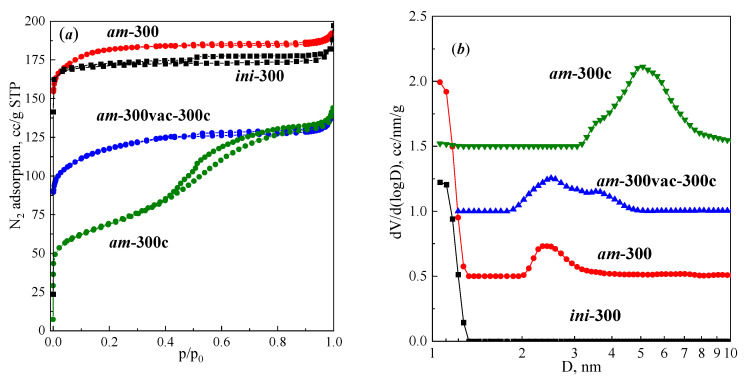
(**a**) N_2_ (−196 °C) adsorption isotherms for ***ini*-300**, ***am*-300**, ***am*-300vac-300c** and ***am*-300c** samples; (**b**) corresponding pore size distributions (NLDFT, adsorption branch).

**Figure 5 materials-16-02308-f005:**
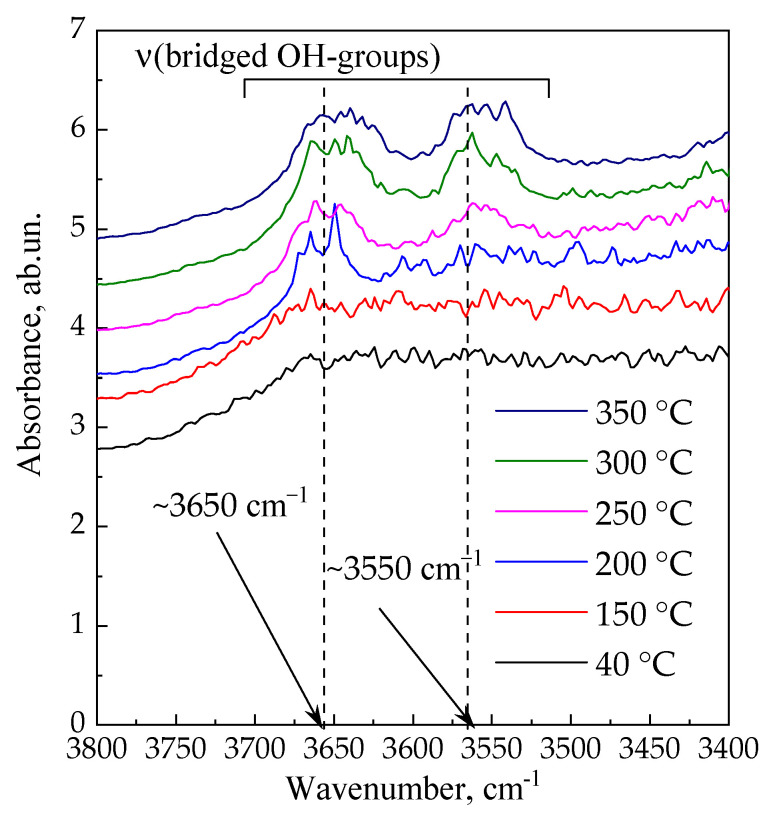
In situ IR-spectra of the ***am*** sample during its heating from room temperature to 350 °C.

**Figure 6 materials-16-02308-f006:**
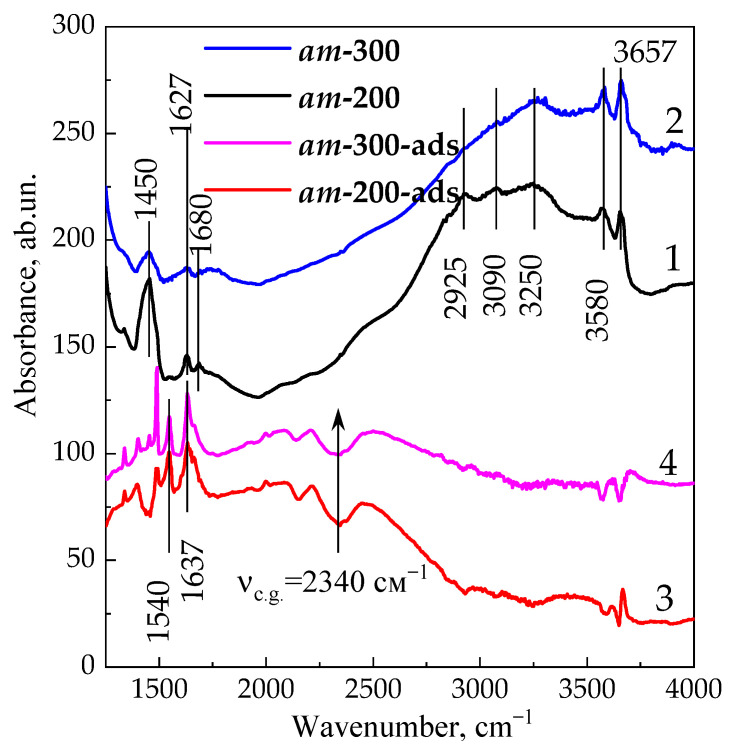
The IR spectra of samples ***am*-200** (Curve 1) and ***am*-300** (Curve 2), and the difference IR spectra obtained by subtracting Spectra 1 and 2 from the spectra taken after pyridine adsorption at 150 ° C, Curves 3 (***am*-200-ads**) and 4 (***am*-300-ads**).

**Figure 7 materials-16-02308-f007:**
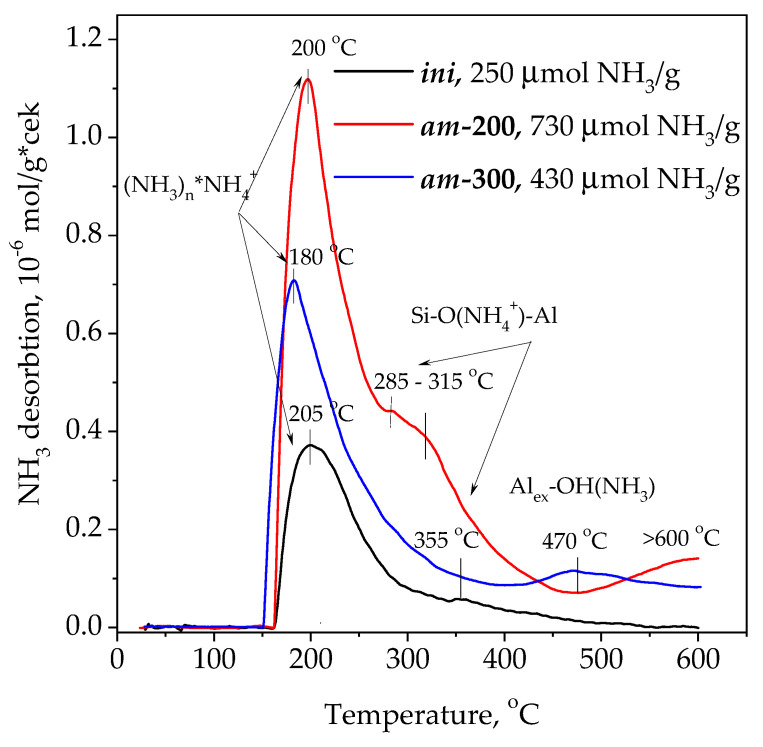
NH_3_ TPD profiles of the samples ***ini*-200**, ***am*-200** and ***am*-300**.

**Figure 8 materials-16-02308-f008:**
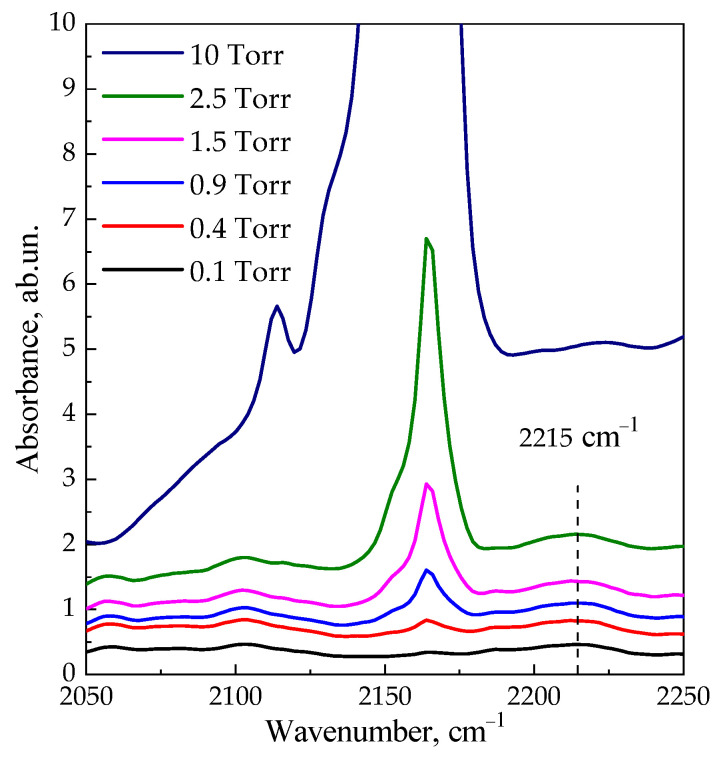
IR spectra of CO adsorbed on the sample ***am*-300** at 77 K and p ~ 0.1, 0.4, 0.9, 1.5, 2.5 and 10 torr.

**Figure 9 materials-16-02308-f009:**
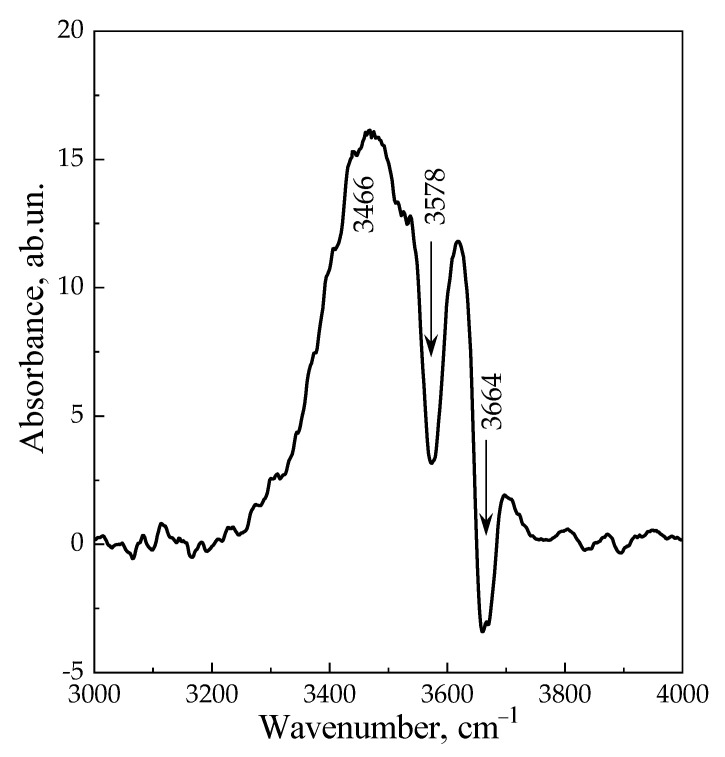
The difference spectrum obtained by subtracting the spectrum after adsorption of CO on the ***am*-300** sample at 77 K and p = 10 Torr (Figure 8) from Curve 2 in Figure 6.

**Figure 10 materials-16-02308-f010:**
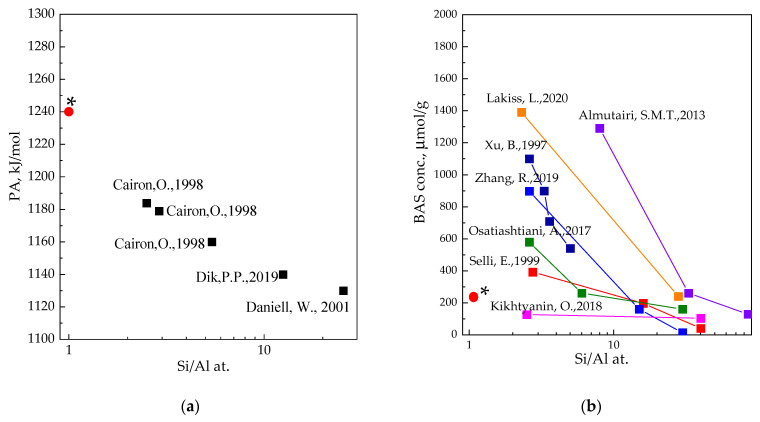
Influence of Si/Al ratio on: (**a**) PA (kJ/mol) of BASs [110,111,112]; (**b**) BAS concentration (μmol/g) [22,27,113,114,115,116,117]. Asterisks designate data from this work.

**Figure 11 materials-16-02308-f011:**
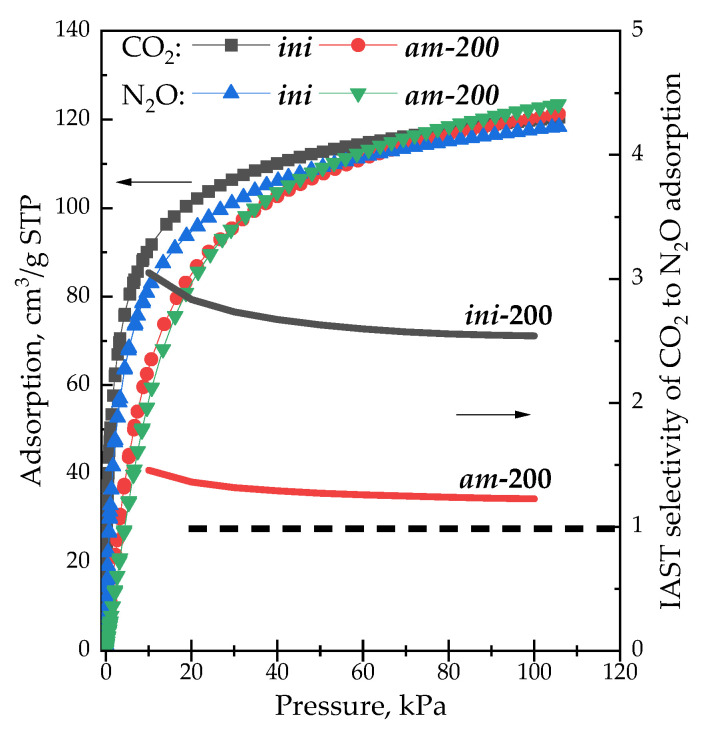
Adsorption isotherms of pure CO_2_ and N_2_O at 0 °C and the corresponding adsorption IAST selectivity for samples ***ini*-200** and ***am*-200**.

**Table 1 materials-16-02308-t001:** Unit cell composition of the samples ^1^.

Sample	K	Na	Al	Si	Si/Al	NH_4_ ^2^
** *ini* **	24	72	96	96	1	0
** *am* **	12	32	96	96	1	52

^1^ measurement error of chemical composition < 5%. ^2^ the content of ammonium ions is determined from the ion balance.

**Table 2 materials-16-02308-t002:** Textural parameters for obtained samples.

Sample	A_s_, m^2^/g	V_mic_, cm^3^/g	V_meso_, cm^3^/g	V_Ʃ_, cm^3^/g	D, nm
***ini*-200**	697	0.27	0	0.27	1.1
***ini*-300**	707	0.28	0	0.28	1.1
***am*-200**	787	0.29	0.03	0.34	1.2; 2.5
***am*-300**	738	0.27	0.05	0.32	1.2; 2.5
***am*-300vac-300c**	440	0.15	0.07	0.21	2.5
***am*-300c**	245	0.05	0.15	0.21	5

## Data Availability

The experimental data were collected using the equipment of the Center for Collective Use “National Center for Research of Catalysts” (CCP “NCIC”) at the Federal Research Center “G.K. Boreskov Institute of Catalysis of the Siberian Branch of the Russian Academy of Sciences”.

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
