# Peer review of "Unusual Acid Sites in LSX Zeolite: Formation Features and Physico-Chemical Properties"

_materials, 2023, doi:10.3390/ma16062308_

Round 1

Reviewer 1 Report

-Introduction section - the ratio described "Si/Al ~1.1 at." should be shown in molar ratio or SAR ratio.

-"Firstly, acid sites of assumedly high concentration are accessible to relatively large substrate molecules" The authors should mention at least a range of substrate molecules size.

- Why does the NH3 TPD present such a strange baseline? Was made some type of treatment in the signal? Typically these signals present baselines that do not return to zero at high temperatures. I suggest the authors present their raw data instead the supposedly treated ones. 

-The duration of each treatment should be inserted in Figure 1.

- (g/g) should be replaced by (wt./wt.)

- Figure 3 - the authors should shift the axis of the isotherms to show the low range of pressure adsorption characteristic of microporous materials (e.g. start the X at -0.01 )

- The authors did not mention why the hysteresis loops are present in Figure 3.

- Is it correct to mention the surface area for microporous materials? (Table 2

- Figure 5 is too noisy. I suggest focusing only in the desired region (i.e. OH ) 

Author Response

We thank a lot the Reviewer for the comprehensive consideration of our manuscript. Please, find below our answers to the provided comments and suggestions.

 Q1: Introduction section - the ratio described "Si/Al ~1.1 at." should be shown in molar ratio or SAR ratio.

A1: Molar Si/Al ratio ~1.1 is in the text now.

Q2: -"Firstly, acid sites of assumedly high concentration are accessible to relatively large substrate molecules" The authors should mention at least a range of substrate molecules size.

A2: Corrected according to the comment.

Q3: Why does the NH3 TPD present such a strange baseline? Was made some type of treatment in the signal? Typically these signals present baselines that do not return to zero at high temperatures. I suggest the authors present their raw data instead the supposedly treated ones.

A3: Thanks for the comment. Indeed, we removed the high-temperature desorption region (above 500 ºC) of ammonia from the NH3 TPD profiles of the samples. This area is not related to the acidity of the sample. It is caused by the amination of extra-framework Al-OH and terminal Si-OH sites during the NH3 TPD experiments. In accordance with the reviewer's comment, Figure 6 has been replaced.

Q4: The duration of each treatment should be inserted in Figure 1.

A4: The duration was inserted.

Q5: (g/g) should be replaced by (wt./wt.)

A5: The designation was replaced

Q6: Figure 3 - the authors should shift the axis of the isotherms to show the low range of pressure adsorption characteristic of microporous materials (e.g. start the X at -0.01 )

A6: The correction was made in accordance with the comment.

Q7: The authors did not mention why the hysteresis loops are present in Figure 3.

A7: The hysteresis loops on the isotherms in Figure 3 are insignificant and very typical to microporous materials. We do not feel the necessity of their special discussion in the text descripting Figure 3.

Q8: Is it correct to mention the surface area for microporous materials? (Table 2)

A8: IUPAC allows such mentioning in the terms of “fingerprint” for microporous materials (see, e.g. Thommes M. et al. Physisorption of gases, with special reference to the evaluation of surface area and pore size distribution (IUPAC Technical Report) // Pure and Applied Chemistry. 2015. Vol. 87, â„– 9–10. P. 1051–1069.). In the particular case of faujazites the correctly applied BET method (with Rouquerol’s consistency criteria, or in the MA-BET form, as it was done in this work) gives the values of the specific surface area close to the accessible geometric area (Bae Y.-S., Yazaydın A.Ö., Snurr R.Q. Evaluation of the BET Method for Determining Surface Areas of MOFs and Zeolites that Contain Ultra-Micropores // Langmuir. 2010. Vol. 26, â„– 8. P. 5475–5483.). Thus, the answer is rather “yes”.

Q9: Figure 5 is too noisy. I suggest focusing only in the desired region (i.e. OH )

A9: We applied the corresponding scaling.

Reviewer 2 Report

In this work, entitled “Unusual acid sites in LSX zeolite: formation features and physico-chemical properties”, Leonova et al. investigate about a method to prepare NH4+-LSX zeolites and its subsequent decationization to form the protonated form of these zeolites maintaining the zeolites crystallinity. Nature and concentration of acid sites are characterized by conventional techniques: NH3 TPD, in-situ IR spectroscopy and IR spectroscopy of probe molecules (pyridine and CO). Modification of the physico-chemical properties of low Si/Al zeolites is and interesting field of study. Modification of the basic/acid properties of X zeolites could be very interesting in order to prepare new multifunctional catalysis and could open their use in new applications.

Authors demonstrate that it is possible to prepare well crystallized ammonium LSX zeolites and the point out that thermal decomposition of ammonia cations give rise to BASs. Finally, they compare the results attained in this work with that attained by other researchers.

The paper is well-structured, easily readable, and could be a nice piece of work. The possibility of synthesizing NH4+ - LSX zeolites is demonstrated. However, although the authors have done an interesting characterization of the samples results could be better organized and discussed. Differences and comparation of the two main samples of the work (am-200 and am-300) between them and with the reference materials (ini-200 and ini-300) should be done in a more ordered manner and comparation should be done clearer and in a more systematic way. Moreover, authors should indicate more clearly in the abstract, results and discussion and conclusions, the unusual characteristics of the acid sites generated in the LSX zeolites by them prepared (as they point out in the tittle). Finally, the advantages of using this type of materials in any application (any traditional field of catalysis, adsorption or ion exchange, or others) is not demonstrated.

Other specific comments:

The title indicates that authors have prepared LSX zeolites with unusual acid sites but in the abstract there is no information about the uncommon acidity of the prepared zeolites. Authors should highlight which are the unusual characteristics of the prepared materials in the abstract. Is it the acidity of 3 BASs per unit cell? Is the strength of the acid sites generated? Both? Is the first time this acid sites concentration and strength are attained in this type of zeolites?

In spite of the elevated number of cites (112 previous works are cited, 40 of them in the Introduction), contextualization about the controlled decationization over X zeolites should be improved. More information about previous results, methods, achievements, limitations should be mentioned and described in the text. This is important to compare and differentiate the results attained by the authors of this work and results obtained by other researchers.

Material and Methods. Regarding ammonia TPD experiments, in the section Materials and Methods (lines 97-98), it is mentioned that samples were pretreated in He flow for 2 hours to remove adsorbed water, followed by cooling down to 75ºC. Which was the temperature employed in the pretreatment? Does this temperature compromise the thermal stability of the samples? On the other hand, as usual, NH3-TPD curves were recorded from 25 to 600 ºC (lines 102-104). Could authors ensure that all the samples tested are stable in the whole temperature range of ammonia TPD experiments?

Results. Line 209-213. Some affirmations over am-300c sample are also applicable to am-300vac-300c samples.

Results. NH3 TPD. Line 230. It is mentioned that “The concentration and thermal stability of BASs were estimated from NH3 TPD”. NH3 TPD is commonly employed in the characterization of acid properties of zeolites (acidity and strength) but this technique is not useful for differentiate between the type of acid sites (BAS or LAS). Redaction and figure 6 should be revised.

As authors indicate, the sample ini-200 is used as a reference material, with no acid sites. In this sample a significant maximum at about 200ºC is observed, corresponding, as authors indicate, to ammonia interacting with residual adsorbed water in the zeolite pores. However, the loss of ammonia for this no acid zeolite extends from 160 to 500 ºC, could authors explain the reason for it?

The maximum at about 200ºC is much important in the am-300 and am-200 samples, is it due to a higher amount of water adsorbed in zeolite pores? Is it due to other phenomena?

Which is the meaning of the temperature of 355 ºC indicated in figure 6? It seems as important for am-300 sample as for ini-200 one.

The sample am-200 possess initially a higher amount of NH4+ than the am-300 one. Due to the different thermal treatment of both samples the ammonium decomposition of the latter sample was more important. Has this fact some influence in the ammonia TPD experiments?

In conclusion, differences between the ammonia desorption profile of am-200 and am-300 should be discussed with more detail (differences in the maxima at about 200, 300 and 500 ºC). Differences in the concentration of acid sites should be also discussed. Moreover, concentration and strength of BASs and LASs should be determined by FTIR with adsorbed probe molecules at different desorption temperature better than by NH3 TPD.

Conclusions. Lines 340-343. Authors conclude “… the adsorption capacity decreases due to the decrease of the number of groups -Si-O(H)-Al- (BAS), accompanied with the formation of an amorphous phase consisting of LASs and extraframework -Si-OH and -Al-OH species.” The formation of this amorphous phase has not been discussed in the Results and discussion section. This fact should be better explained and discussed in the corresponding section.

No mention to the unusual acid sites in LSX zeolite indicated in the Title is found in the conclusions section.

Finally, please revise the following typos and grammar inconsistences detected:

o   Line 163-164: Please, redaction should be revised.

o   Line 201: Sample ini-200 is mentioned while it should be ini-300.

o   Line 304-305 and 316: “…from curve 2 in Figure 8…” Probably, authors refer to curve 2 in Figure 7.

Author Response

Thank you very much for useful comments and notes! Please find our answers in the attached file.

Round 2

Reviewer 2 Report

As the comments raised by the reviewers were mostly implemented in the revised manuscript, the manuscript can be accepted for publication.